# TAB-VCR: Tags and Attributes based Visual Commonsense Reasoning Baselines

**Jingxiang Lin, Unnat Jain, Alexander G. Schwing**
University of Illinois at Urbana-Champaign
`https://deanplayerljx.github.io/tabvcr`

## Abstract

Reasoning is an important ability that we learn from a very early age. Yet, reasoning is extremely hard for algorithms. Despite impressive recent progress that has been reported on tasks that necessitate reasoning, such as visual question answering and visual dialog, models often exploit biases in datasets. To develop models with better reasoning abilities, recently, the new visual commonsense reasoning (VCR) task has been introduced. Not only do models have to answer questions, but also do they have to provide a reason for the given answer. The proposed baseline achieved compelling results, leveraging a meticulously designed model composed of LSTM modules and attention nets. Here we show that a much simpler model obtained by ablating and pruning the existing intricate baseline can perform better with half the number of trainable parameters. By associating visual features with attribute information and better text to image grounding, we obtain further improvements for our simpler & effective baseline, **TAB-VCR**. We show that this approach results in a 5.3%, 4.4% and 6.5% absolute improvement over the previous state-of-the-art [103] on question answering, answer justification and holistic VCR.

## 1 Introduction

Reasoning abilities are important for many tasks such as answering of (referential) questions, discussion of concerns and participation in debates. While we are trained to ask and answer "why" questions from an early age and while we generally master answering of questions about observations with ease, visual reasoning abilities are all but simple for algorithms.

Nevertheless, respectable accuracies have been achieved recently for many tasks where visual reasoning abilities are necessary. For instance, for visual question answering [9, 32] and visual dialog [20], compelling results have been reported in recent years, and many present-day models achieve accuracies well beyond random guessing on challenging datasets such as [30, 47, 109, 37]. However, it is also known that algorithm results are not stable at all and trained models often leverage biases to answer questions. For example, both questions about the existence and non-existence of a "pink elephant" are likely answered affirmatively, while questions about counting are most likely answered with the number 2. Even more importantly, a random answer is returned if the model is asked to explain the reason for the provided answer.

To address this concern, a new challenge on "visual commonsense reasoning" [103] was introduced recently, combining reasoning about physics [69, 99], social interactions [2, 89, 16, 33], understanding of procedures [107, 3] and forecasting of actions in videos [84, 26, 108, 90, 28, 74, 100]. In addition to answering a question about a given image, the algorithm is tasked to provide a rationale to justify the given answer. In this new dataset the questions, answers, and rationales are expressed using a natural language containing references to the objects. The proposed model, which achieves compelling results, leverages those cues by combining a long-short-term-memory (LSTM) module based deep net with attention over objects to obtain grounding and context.

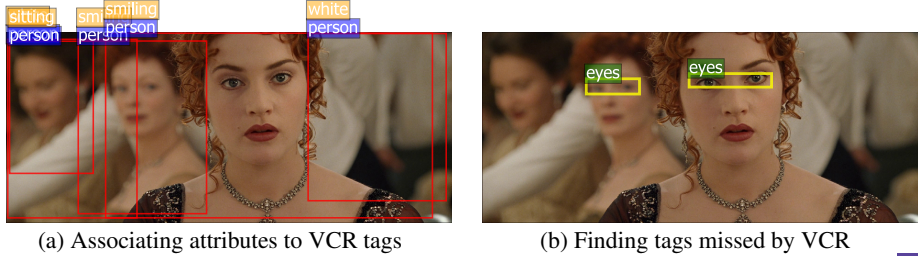

(a) Associating attributes to VCR tags　　　　　　(b) Finding tags missed by VCR

Figure 1: **Motivation and improvements**. (a) The VCR object detections, *i.e.*, red boxes and labels in blue are shown. We capture visual attributes by replacing the image classification CNN (used in previous models) with an image+attribute classification CNN. The predictions of this CNN are highlighted in orange. (b) Additionally, many nouns referred to in the VCR text aren't *tagged*, *i.e.* grounded to objects in the image. We utilize the same image CNN as (a) to detect objects and ground them to text. The *new tags* we found augment the VCR tags, and are highlighted with yellow bounding boxes and the associated labels in green.

However, the proposed model is also very intricate. In this paper we revisit this baseline and show that a much simpler model with less than half the trainable parameters achieves significantly better results. As illustrated in Fig. 1, different from existing models, we also show that attribute information about objects and careful detection of objects can greatly improve the model performance. To this end we extract visual features using an image CNN trained for the auxillary task of attribute prediction. In addition to encoding the image, we utilize the CNN to augment the object-word groundings provided in the VCR dataset. An effective grounding for these *new tags* is obtained by using a combination of part-of-speech tagging and Wu Palmer similarity. We refer to our developed tagging and attribute baseline as **TAB-VCR**.

We evaluate the proposed approach on the challenging and recently introduced visual commonsense reasoning (VCR) dataset [103]. We show that a simple baseline which carefully leverages attribute information and object detections is able to outperform the existing state-of-the-art by a large margin despite having less than half the trainable model parameters.

## 2　Related work

In the following we briefly discuss work related to vision based question answering, explainability and visual attributes.

**Visual Question Answering.** Image based question answering has continuously evolved in recent years, particularly also due to the release of various datasets [65, 73, 9, 101, 30, 104, 109, 47, 44, 72, 71]. Specifically, Zhang et al. [104] and Goyal et al. [32] focus on balancing the language priors of Antol et al. [9] for abstract and real images. Agrawal et al. [1] take away the IID assumption to create different distributions of answers for train and test splits, which further discourages transfer of language priors. Hudson and Manning [37] balance open questions in addition to binary questions (as in Goyal et al. [32]). Image based dialog [20, 24, 21, 42, 60] can also be posed as a step by step image based question answering and question generation [68, 43, 55] problem. Similarly related are question answering datasets built on videos [86, 64, 51, 52] and those based on visual embodied agents [31, 22].

Various models have been proposed for these tasks, particularly for VQA [9, 32], selecting sub-regions of an image [87], single attention [13, 98, 6, 19, 29, 82, 97, 39, 102], multimodal attention [59, 79, 70], memory nets and knowledge bases [96, 94, 91, 62], improvements in neural architecture [66, 63, 7, 8] and bilinear pooling representations [29, 46, 12].

**Explainability.** The effect of explanations on learning have been well studied in Cognitive Science and Psychology [57, 92, 93]. Explanations play a critical role in child development [50, 18] and more generally in educational environments [15, 76, 77]. Explanation based models for applications in medicine & tutoring have been previously proposed [83, 88, 49, 17]. Inspired by these findings, language and vision research on attention mechanism help to provide insights into decisions made by deep net models [59, 80]. Moreover, explainability in deep models has been investigated by modifying CNNs to focus on object parts [106, 105], decomposing questions using neural modular substructures [8, 7, 23], and interpretable hidden units in deep models [10, 11]. Most relevant to our research are works on natural language explanations. This includes multimodal explanation [38] and textual explanations for classifier decisions [35] and self driving vehicles [45].

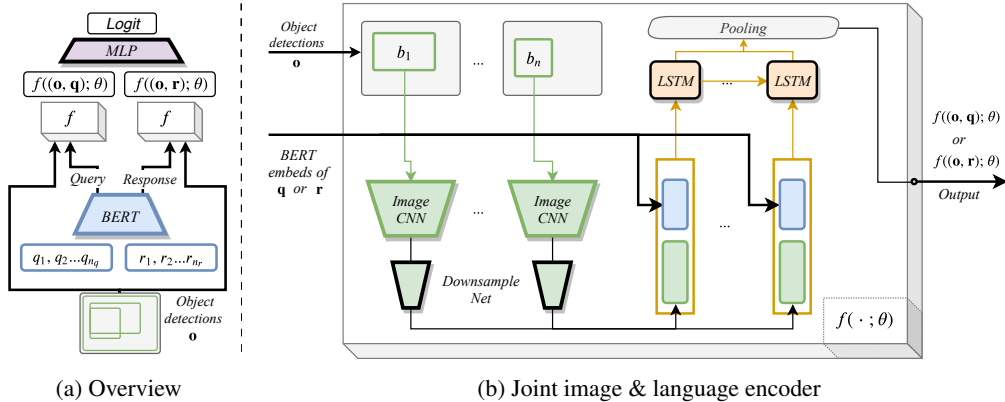

<center>(a) Overview            (b) Joint image & language encoder</center>

Figure 2: (a) **Overview of the proposed TAB-VCR model**: Inputs are the image (with object bounding boxes), a query and a candidate response. Sentences (query & response) are represented using BERT embeddings and encoded jointly with the image using a deep net module $f(\cdot; \theta)$. The representations of query and response are concatenated and scored via a multi-layer perceptron (MLP); (b) **Details of joint image & language encoder** $f(\cdot; \theta)$: BERT embeddings of each word are concatenated with their corresponding local image representation. This information is pass through an LSTM and pooled to give the output $f((I, \mathbf{w}); \theta)$. The network components outlined in black, *i.e.*, MLP, downsample net and LSTM are the only components with trainable parameters.

**Visual Commonsense Reasoning.** The recently introduced Visual Commonsense Reasoning dataset [103] combines the above two research areas, studying explainability (reasoning) through two multiple-choice subtasks. First, the question answering subtask requires to predict the answer to a challenging question given an image. Second, and more connected to explainability, is the answer justification subtask, which requires to predict the rationale given a question and a correct answer. To solve the VCR task, Zellers et al. [103] base their model on a convolutional neural network (CNN) trained for classification. Instead, we associate VCR detections with visual attribute information to obtain significant improvements with no architectural change or additional parameter cost. We discuss related work on visual attributes in the following.

**Visual attributes.** Attributes are semantic properties to describe a localized object. Visual attributes are helpful to describe an unfamiliar object category [27, 48, 78]. Visual Genome [47] provides over 100k images along with their scene graphs and attributes. Anderson et al. [5] capture attributes in visual features by using an auxiliary attribute prediction task on a ResNet101 [34] backbone.

## 3 Attribute-based Visual Commonsense Reasoning

We are interested in visual commonsense reasoning (VCR). Specifically, we study simple yet effective models and incorporate important information missed by previous methods – attributes and additional object-text groundings. Given an input image, the VCR task is divided into two subtasks: (1) **question answering** ($Q{\rightarrow}A$): given a question (Q), select the correct answer (A) from four candidate answers; (2) **answer justification** ($QA{\rightarrow}R$): given a question (Q) and its correct answer (A), select the correct rationale (R) from four candidate rationales. Importantly, both subtasks can be unified: choosing a *response* from four options given a *query*. For $Q{\rightarrow}A$, the query is a question and the options are candidate answers. For $QA{\rightarrow}R$, the query is a question appended by its correct answer and the options are candidate rationales. Note, the $Q{\rightarrow}AR$ task combines both, *i.e.*, a model needs to succeed at both $Q{\rightarrow}A$ and $QA{\rightarrow}R$. The proposed method focuses on choosing a response given a query, for which we introduce notation next.

We are given an *image*, a *query*, and four candidate *responses*. The words in the query and responses are grounded to objects in the image. The query and response are collections of words, while the image data is a collection of object detections. One of the detections also corresponds to the entire image, symbolizing a global representation. The image data is denoted by the set $\mathbf{o} = (o_i)_{i=1}^{n_o}$, where each $o_i$, $i \in \{1, \ldots, n_o\}$, consists of a bounding box $b_i$ and a class label $l_i \in \mathcal{L}$[1]. The query is composed of a sequence $\mathbf{q} = (q_i)_{i=1}^{n_q}$, where each $q_i$, $i \in \{1, \ldots, n_q\}$, is either a word in the vocabulary $\mathcal{V}$ or a tag referring to a bounding box in $\mathbf{o}$. A data point consists of four responses and

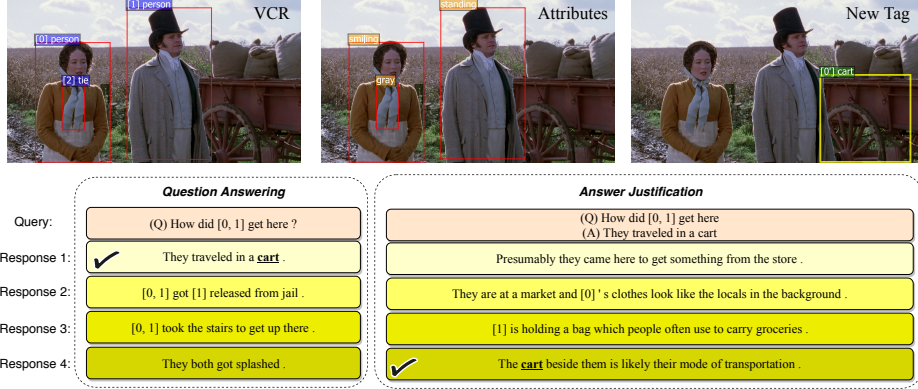

(a) Direct match of word **cart** (in text) and the same label (in image).

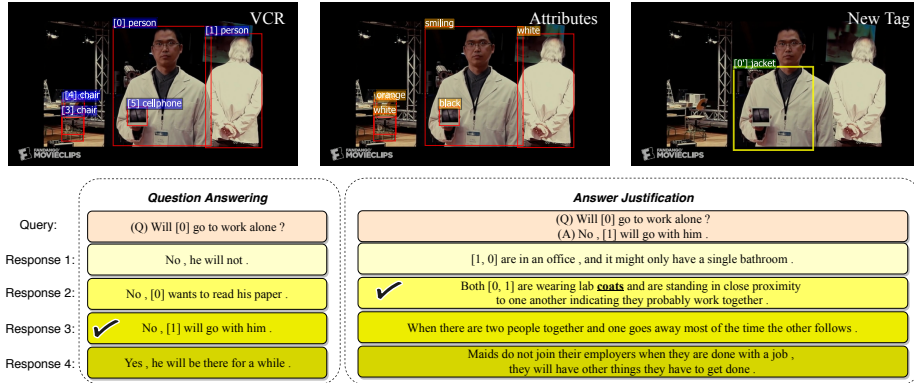

(b) Word sense based match of word **coats** and label 'jacket' with the same meaning.

Figure 3: **Qualitative results:** Two types of *new tags* found by our method are (a) direct matches and (b) word sense based matches. Note that the images on the left show the object detections provided by VCR. The images in the middle show the attributes predicted by our model and thereby captured in visual features. The images on the right show *new tags* detected by our proposed method. Below the images are the question answering and answer justification subtasks.

we denote a response by the sequence $\mathbf{r} = (r_i)_{i=1}^{n_r}$, where $r_i$, $i \in \{1, \ldots, n_r\}$, (like the query) can either refer to a word in the vocabulary $\mathcal{V}$ or a tag.

We develop a conceptually simple joint encoder for language and image information, $f(\,\cdot\,;\theta)$, where $\theta$ is the catch-all for all the trainable parameters.

In the remainder of this section, we first present an overview of our approach. Subsequently, we discuss details of the joint encoder $f(\,\cdot\,;\theta)$. Afterward, we introduce how to incorporate attribute information and find *new tags*, which helps improve the performance of our simple baseline. We defer details about training and implementation to the supplementary material.

## 3.1 Overview

As mentioned, visual commonsense reasoning requires to choose a response from four candidates. Here, we score each candidate separately. The separate scoring of responses is necessary to build a more widely applicable framework, which is independent of the number of responses to be scored.

Our proposed approach is outlined in Fig. 2(a). The three major components of our approach are: (1) BERT [25] embeddings for words; (2) a joint encoder $f(\,\cdot\,;\theta)$ to obtain $(\mathbf{o}, \mathbf{q})$ and $(\mathbf{o}, \mathbf{r})$ representations; and (3) a multi-layer perceptron (MLP) to score these representations. Each word in the query set $\mathbf{q}$ and response set $\mathbf{r}$ is embedded via BERT. The BERT embeddings of $\mathbf{q}$ and associated image data from $\mathbf{o}$ are jointly encoded to obtain the representation $f((\mathbf{o}, \mathbf{q}); \theta)$. An analogous representation for responses is obtained via $f((\mathbf{o}, \mathbf{r}); \theta)$. Note that the joint encoder is identical for both the query and the response. The two representations are concatenated and scored via an MLP. These scores or logits are further normalized using a softmax. The network is trained end-to-end using a cross-entropy loss of predicted probabilities vis-à-vis correct responses.

---
**Algorithm 1** Finding *new tags*

---
1: Forward pass through image CNN to obtain object detections $\hat{\mathbf{o}}$
2: $\hat{\mathcal{L}} \leftarrow$ set(all class labels in $\hat{\mathbf{o}}$)
3: **for** $w \in \mathbf{w}$ where $\mathbf{w} \in \{\mathbf{q}, \mathbf{r}\}$ **do**
4:     **if** $w$ is tag **then** $w \leftarrow$ remap($w$)
5: new_tags $\leftarrow \{\}$
6: **for** $w \in \mathbf{w}$ where $\mathbf{w} \in \{\mathbf{q}, \mathbf{r}\}$ **do**
7:     **if** (pos_tag($w|\mathbf{w}$) $\in \{$NN, NNS$\}$) and (wsd_synset($w, \mathbf{w}$) has a noun) **then**
8:         **if** $w \in \hat{\mathcal{L}}$ **then**               ▷ Direct match between word and detections
9:             new_detections $\leftarrow$ detections in $\hat{\mathbf{o}}$ corresponding to $w$
10:             add ($w$, new_detections) to new_tags
11:         **else**                    ▷ Use word sense to match word and detections
12:             max_wup $\leftarrow 0$
13:             word_lemma $\leftarrow$ lemma($w$)
14:             word_sense $\leftarrow$ first_synset(word_lemma)
15:             **for** $\hat{l} \in \hat{\mathcal{L}}$ **do**
16:                 **if** wup_similarity(first_synset($\hat{l}$), word_sense) $>$ max_wup **then**
17:                     max_wup $\leftarrow$ wup_similarity(first_synset($\hat{l}$), word_sense)
18:                     best_label $\leftarrow \hat{l}$
19:             **if** max_wup $> k$ **then**
20:                 new_detections $\leftarrow$ detections in $\hat{\mathbf{o}}$ corresponding to best_label
21:                 add ($w$, new_detections) to new_tags

---

Next, we provide details of the joint encoder before we describe our approach to incorporate attributes and better image-text grounding, to improve the performance.

### 3.2 Joint image & language encoder

The joint language and image encoder is illustrated in Fig. 2(b). The inputs to the joint encoder are word embeddings of a sentence (either $\mathbf{q}$ or $\mathbf{r}$) and associated object detections from $\mathbf{o}$. The local image region defined by these bounding boxes is encoded via an image CNN to a $2048$ dimensional vector. This vector is projected to a $512$ dimensional embedding, using a fully connected *downsample net*. The language and image embeddings are concatenated and transformed using a long-short term memory network (LSTM) [36]. Note that for non-*tag* words, *i.e.*, words without an associated object detection, the object detection corresponding to the entire image is utilized. The outputs of each unit of the LSTM are pooled together to obtain the final joint encoding of $\mathbf{q}$ (or $\mathbf{r}$) and $\mathbf{o}$. Note that the network components with a ⎣black outline,⎦ *i.e.*, the downsample net and LSTM are the only components with trainable parameters. We design this so that no gradients need to be propagated back to the image CNN or to the BERT model, since both of them are parameter intensive, requiring significant training time and data. This choice facilitates the pre-computation of language and image features for faster training and inference.

### 3.3 Improving visual representation & image-text grounding

**Attributes capturing visual features.** Almost all previous VCR baselines have used a CNN trained for ImageNet classification to extract visual features. Note that the class label $l_i$ for each bounding box is already available in the dataset and incorporated in the models (previous and ours) via BERT embeddings. We hypothesize that visual question answering and reasoning benefits from information about object characteristics and attributes. This intuition is illustrated in Fig. 3 where attributes add valuable information to help reason about the scene, such as '*black* picture,' '*gray* tie,' and '*standing* man.' To validate this hypothesis we deploy a pretrained attribute classifier which augments every detected bounding box $b_i$ with a set of attributes such as colors, texture, size, and emotions. We show the attributes predicted by our model's image CNN in Fig. 1(a). For this, we take advantage of work by Anderson et al. [5] as it incorporates attribute features to improve performance on language and vision tasks. Note that Zellers et al. [103] evaluate the model proposed by Anderson et al. [5] with BERT embeddings to obtain $39.6\%$ accuracy on the test set of the $Q{\rightarrow}AR$ task. As detailed in Sec. 4.3, with the same CNN and BERT embeddings, our network achieves $50.5\%$. We achieve this by capturing recurrent information of LSTM modules via pooling and better scoring through an

MLP. This is in contrast to Zellers et al. [103], where the VQA 1000-way classification is removed and the response representation is scored using a dot product.

***New tags* for better text to image grounding.** Associating a word in the text with an object detection in the image, *i.e.*, $o_i = (b_i, l_i)$ is what we commonly refer to as text-image grounding. Any word serving as a pointer to a detection is referred to as a *tag* by Zellers et al. [103]. Importantly, many nouns in the text (query or responses) aren't grounded with their appearance in the image. We explain possible reasons in Sec. 4.4. To overcome this shortcoming, we develop Algorithm 1 to find new text-image groundings or *new tags*. A qualitative example is illustrated in Fig. 3. Nouns such as 'cart' and 'coats' weren't tagged by VCR, while our TAB-VCR model can tag them.

Specifically, for text-image grounding we first find detections $\hat{\mathbf{o}}$ (in addition to VCR provided $\mathbf{o}$) using the image CNN. The set of unique class labels in $\hat{\mathbf{o}}$ is assigned to $\hat{\mathcal{L}}$. Both $\mathbf{q}$ and $\mathbf{r}$ are modified such that all *tags* (pointers to detections in the image) are remapped to natural language (class label of the detection). This is done via the `remap` function. We follow Zellers et al. [103] and associate a gender neutral name for the 'person' class. For instance, "How did [0,1] get here?" in Fig. 3 is remapped to "How did Adrian and Casey get here?". This remapping is necessary for the next step of the part-of-speech (POS) tagging which operates only on natural language.

Next, the POS tagging function (`pos_tag`) parses a sentence $\mathbf{w}$ and assigns POS tags to each word $w$. For finding *new tags*, we are only interested in words with the POS tag being either singular noun (NN) or plural noun (NNS). For these noun words, we check if a word $w$ directly matches a label in $\hat{\mathcal{L}}$. If such a direct match exists, we associate $w$ to the detections of the matching label. As shown in Fig. 3(a), this direct matching associates the word **cart** in the text (response 1 of the $Q{\rightarrow}A$ subtask and response 4 of the $QA{\rightarrow}R$ subtask) to the detection corresponding to label 'cart' in the image, creating a *new tag*.

If there is no such direct match for $w$, we find matches based on word sense. This is motivated in Fig. 3(b) where the word 'coat' has no direct match to any image label in $\hat{\mathcal{L}}$. Rather there is a detection of 'jacket' in the image. Notably, the word 'coat' has multiple word senses, such as 'an outer garment that has sleeves and covers the body from shoulder down' and 'growth of hair or wool or fur covering the body of an animal.' Also, 'jacket' has multiple word senses, two of which are 'a short coat' and 'the outer skin of a potato'. As can be seen, the first word senses of 'coat' and 'jacket' are similar and would help match 'coat' to 'jacket.' Having said that, the second word senses are different from common use and from each other. Hence, for words that do not directly match a label in $\hat{\mathcal{L}}$, choosing the appropriate word sense is necessary. To this end, we adopt a simple approach, where we use the most frequently used word sense of $w$ and of labels in $\hat{\mathcal{L}}$. This is obtained using the first synset in Wordnet in NLTK [67, 58]. Then, using the first synset of $w$ and labels in $\hat{\mathcal{L}}$, we find the best matching label 'best_label' corresponding to the highest Wu-Palmer similarity between synsets [95]. Additionally, we lemmatize $w$ before obtaining its first synset. If the Wu-Palmer similarity between word $w$ and the 'best_label' is greater than a threshold $k$, we associate the word to the detections of 'best_label.' Overall this procedure leads to *new tags* where text and label aren't the same but have the same meaning. We found $k = 0.95$ was apt for our experiments. While inspecting, we found this algorithm missed to match the word 'men' in the text to the detection label 'man.' This is due to the 'lemmatize' function provided by NLTK [58]. Consequently, we additionally allow *new tags* corresponding to this 'men-man' match.

This algorithm permits to find *new tags* in $7.1\%$ answers and $32.26\%$ rationales. A split over correct and incorrect responses is illustrated in Fig. 4. These *new tag* detections are used by our *new tag* variant **TAB-VCR**. If there is more than one detection associated with a *new tag*, we average the visual features at the step before the LSTM in the joint encoder.

**Implementation details.** We defer specific details about training, implementation and design choices to the supplementary material. The code can be found at `https://github.com/deanplayerljx/tab-vcr`.

## 4   Experiments

In this section, we first introduce the VCR dataset and describe metrics for evaluation. Afterward, we quantitatively compare our approach and improvements to the current state-of-the-art method [103] and to top VQA models. We include a qualitative evaluation of TAB-VCR and an error analysis.

| | $Q{\to}A$ (val) | $QA{\to}R$ (val) | $Q{\to}AR$ (val) | Params (Mn) (total) | (trainable) |
|---|---|---|---|---|---|
| R2C (Zellers et al. [103]) | 63.8 | 67.2 | 43.1 | 35.3 | 26.8 |
| *Improving R2C* | | | | | |
| R2C + Det-BN | 64.49 | 67.02 | 43.61 | 35.3 | 26.8 |
| R2C + Det-BN + Freeze (R2C++) | 65.30 | 67.55 | 44.41 | 35.3 | 11.7 |
| R2C++ + Resnet101 | 67.55 | 68.35 | 46.42 | 54.2 | 11.7 |
| R2C++ + Resnet101 + Attributes | 68.53 | 70.86 | 48.64 | 54.0 | 11.5 |
| *Ours* | | | | | |
| Base | 66.39 | 69.02 | 46.19 | 28.4 | 4.9 |
| Base + Resnet101 | 67.50 | 69.75 | 47.51 | 47.4 | 4.9 |
| Base + Resnet101 + Attributes | 69.51 | 71.57 | 50.08 | 47.2 | 4.7 |
| Base + Resnet101 + Attributes + New Tags (**TAB-VCR**) | **69.89** | **72.15** | **50.62** | 47.2 | 4.7 |

Table 1: Comparison of our approach to the current state-of-the-art R2C [103] on the validation set. Legend: **Det-BN**: Deterministic testing using train time batch normalization statistics. **Freeze**: Freeze all parameters of the image CNN. **ResNet101**: ResNet101 backbone as image CNN (default is ResNet50). **Attributes**: Attribute capturing visual features by using [5] (which has a ResNet101 backbone) as image CNN. **Base**: Our base model, as detailed in Fig. 2(b) and Sec. 3.1. **New Tags**: Augmenting object detection set with *new tags* (as detailed in Sec. 3.3), *i.e.*, grounding additional nouns in the text to the image.

| Model | $Q{\to}A$ | $QA{\to}R$ | $Q{\to}AR$ |
|---|---|---|---|
| Revisited [41] | 57.5 | 63.5 | 36.8 |
| BottomUp [5] | 62.3 | 63.0 | 39.6 |
| MLB [46] | 61.8 | 65.4 | 40.6 |
| MUTAN [12] | 61.0 | 64.4 | 39.3 |
| R2C [103] | 65.1 | 67.3 | 44.0 |
| **TAB-VCR (ours)** | **70.4** | **71.7** | **50.5** |

Table 2: **Evaluation on test set:** Accuracy on the three VCR tasks. Comparison with top VQA models + BERT performance (source: [103]). Our best model outperforms R2C [103] on the test set by a significant margin.

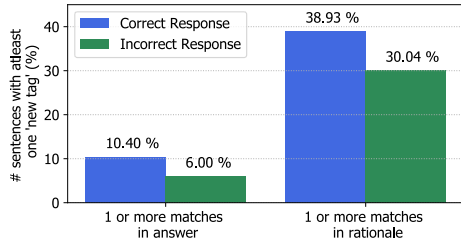

Figure 4: **New tags:** Percentage of response sentences with a *new tag*, *i.e.*, a new grounding for noun and object detection. Correct responses more likely have new detections than incorrect ones.

## 4.1 Dataset

We train our models on the visual commonsense reasoning dataset [103] which contains over 212k (train set), 26k (val set) and 25k (test set) questions on over 110k unique movie scenes. The scenes were selected from LSMDC [75] and MovieClips, after they passed an 'interesting filter.' For each scene, workers were instructed to created 'cognitive-level' questions. Workers answered these questions and gave a reasoning or *rationale* for the answer.

## 4.2 Metrics

Models are evaluated with classification accuracy on the $Q{\to}A$, $QA{\to}R$ subtasks and the holistic $Q{\to}AR$ task. For train and validation splits, the correct labels are available for development. To prevent overfitting, the test set labels were not released. Since evaluation on the test set is a manual effort by Zellers et al. [103], we provide numbers for our best performing model on the test set and illustrate results for the ablation study on the validation set.

## 4.3 Quantitative evaluation

Tab. 1 compares the performance of variants of our approach to the current state-of-the-art R2C [103]. While we report validation accuracy on both subtasks ($Q{\to}A$ and $QA{\to}R$) and the joint ($Q{\to}AR$) task in Tab. 1, in the following discussion we refer to percentages with reference to $Q{\to}AR$.

We make two modifications to improve R2C. The first is `Det-BN` where we calculate and use train time batch normalization [40] statistics. Second, we `freeze` all the weights of the image CNN in R2C, whereas Zellers et al. [103] keep the last block trainable. We provide a detailed study on `freeze` later. With these two minor changes, we obtain an improvement (1.31%) in performance and a significant reduction in trainable parameters (15Mn). We use the shorthand `R2C++` to refer to this improved variant of R2C.

Our `base` model (described in Sec. 3) which includes (`Det-BN`) and `Freeze` improvements, improves over R2C++ by 1.78%, while being conceptually simple, having half the number of trainable parame-

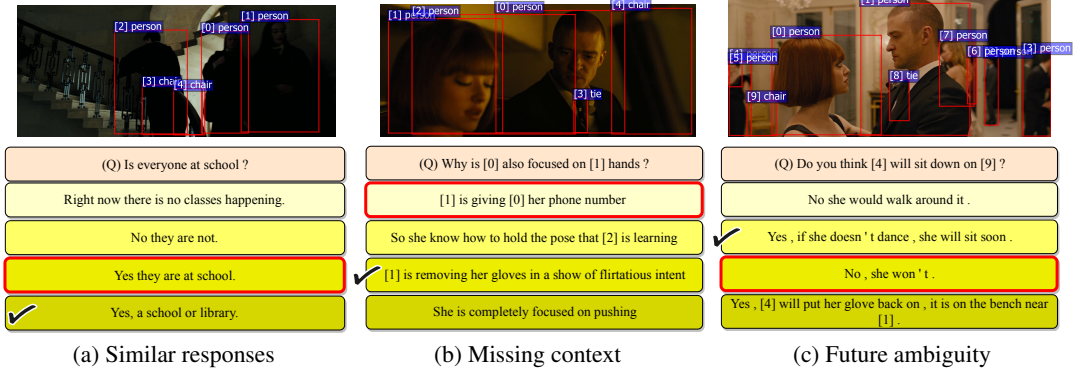

| (a) Similar responses | (b) Missing context | (c) Future ambiguity |

Figure 5: **Qualitative analysis of error modes:** Responses with similar meaning (left), lack of context (middle) or ambiguity in future actions (right). Correct answers are marked with ticks and our model's incorrect prediction is outlined in red.

| Encoder | $Q{\to}A$ | $QA{\to}R$ | $Q{\to}AR$ | Params |
|---|---|---|---|---|
| Shared | 69.89 | 72.15 | 50.62 | 4.7M |
| Unshared | 69.59 | 72.25 | 50.35 | 7.9M |

Table 3: Effect of shared *vs.* unshared parameters in the joint encoder $f(\,\cdot\,;\theta)$ of the TAB-VCR model.

| VCR subtask | Avg. no. of *tags* in query+response | | |
|---|---|---|---|
| | (a) all | (b) correct | (c) errors |
| $Q{\to}A$ | 2.673 | 2.719 | 2.566 |
| $QA{\to}R$ | 4.293 | 4.401 | 4.013 |

Table 4: Error analysis as a function of number of tags. Less image-text grounding increases TAB-VCR errors.

ters. By using a more expressive ResNet as image CNN model (`Base + Resnet101`), we obtain another 1.32% improvement. We obtain another big increase of 2.57% by leveraging attributes capturing visual features (`Base + Resnet101 + Attributes`). Our best performing variant incorporates *new tags* during training and inference (TAB-VCR) with a final 50.62% on the validation set. We ablate `R2C++` with `ResNet101` and `Attributes` modifications, which leads to better performance too. This suggests our improvements aren't confined to our particular net. Additionally, we share the encoder for query and responses. We empirically studied the effect of sharing encoder parameters and found no significant difference (Tab. 3) when using separate weights, which comes at the cost of 3.2M extra trainable parameters. Note that Zellers et al. [103] also share the encoder for query and response processing. Hence, our design choice makes the comparison fair.

In Tab. 2 we show results evaluating the performance of TAB-VCR on the private test set, set aside by Zellers et al. [103]. We obtain a 5.3%, 4.4% and 6.5% absolute improvement over R2C on the test set. We perform much better than top VQA models which were adapted for VCR in [103]. Models evaluated on the test set are posted on the leaderboard[2]. We appear as 'TAB-VCR' and outperform prior peer-reviewed work. At the time of writing (23[rd] May 2019) TAB-VCR ranked second in the single model category. After submission of this work other reports addressing VCR have been released. At the time of submitting this camera-ready (27[th] Oct 2019), TAB-VCR ranked seventh among single models on the leaderboard. Based on the available reports [54, 85, 4, 53, 61, 14], most of these seven methods capture the idea of re-training BERT with extra information from Conceptual Captions [81]. This, in essence, is orthogonal to our *new tags* and attributes approach to build simple and effective baselines with significantly fewer parameters.

Fig. 4 illustrates the effectiveness of our *new tag* detection, where 10.4% correct answers had at least one *new tag* detected. With 38.93%, the number is even higher for correct rationales. This is intuitive as humans refer to more objects while reasoning about an answer than the answer itself.

**Finetuning *vs*. freezing last conv block.** In Tab. 5 we study the effect of finetuning the last conv block of ResNet101 and the downsample net. Zellers et al. [103] use row #1. We assess lower learning rates – 0.5x, 0.25x, and 0.125x (#2 to #4). We chose to freeze the conv block (#5) to reduce trainable parameters by 15M, with slight improvement in performance. By comparing #5 and #6, we find the presence of downsample net to reduce the model size and improve performance. After conducting this ablation study for the `base` model's architecture design, we updated the python dependency packages. This update lead to a slight difference in the accuracy of #5 in Tab. 5 (before the update) and the final accuracy reported in Tab. 1 (after the update). However, the versions of python dependencies are consistent across all variants listed in Tab. 5.

| # | 4th conv block | Downsample net | $Q{\rightarrow}A$ | $QA{\rightarrow}R$ | $Q{\rightarrow}AR$ | Trainable params (mn) |
|---|---|---|---|---|---|---|
| 1 | ■ | ■ | 64.57 | 68.86 | 44.60 | 19.9 |
| 2 | ■ (1/2) | ■ | 64.26 | 68.14 | 44.08 | 19.9 |
| 3 | ■ (1/4) | ■ | 63.11 | 67.73 | 42.87 | 19.9 |
| 4 | ■ (1/8) | ■ | 63.51 | 67.49 | 43.21 | 19.9 |
| 5 | ■ | ■ | 66.47 | 69.22 | 46.45 | 4.9 |
| 6 | ■ | ■ | 65.30 | 69.09 | 45.57 | 7.0 |

Table 5: Ablation for `base` model: ■: Finetuning and ■: Freezing weights of the fourth conv block in ResNet101 image CNN. Presence and absence of downsample net (to project image representation from 2048 to 512) is denoted by ■ and ■.

| Ques. type | Matching patterns | Counts | $Q{\rightarrow}A$ | $QA{\rightarrow}R$ |
|---|---|---|---|---|
| what | what | 10688 | 72.30 | 72.74 |
| why | why | 9395 | 65.14 | 73.02 |
| isn't | is, are, was, were, isn't | 1768 | 75.17 | 67.70 |
| where | where | 1546 | 73.54 | 73.09 |
| how | how | 1350 | 60.67 | 69.19 |
| do | do, did, does | 655 | 72.82 | 65.80 |
| who | who, whom, whose | 556 | 86.69 | 69.78 |
| will | will, would, wouldn't | 307 | 74.92 | 73.29 |

Table 6: Accuracy by question type (with at least 100 counts) of `TAB-VCR` model. *Why* & *how* questions are most challenging for the $Q{\rightarrow}A$ subtask.

### 4.4 Qualitative evaluation and error analysis

We illustrate the qualitative results in Fig. 3. We separate the image input to our model into three parts, for easy visualization. We show VCR detections & labels, attribute prediction of our image CNN and *new tags* in the left, middle and right images. Note how our model can ground important words. For instance, for the example shown in Fig. 3(a), the correct answer and rationale prediction is based on the cart in the image, which we ground. The word **'cart'** wasn't grounded in the original VCR dataset. Similarly, grounding the word **coats** helps to answer and reason about the example in Fig. 3(b).

**Explanation for missed tags.** As discussed in Sec. 3.3, the VCR dataset contains various nouns that aren't tagged such as 'eye,' 'coats' and 'cart' as highlighted in Fig. 1 and Fig. 3. This could be accounted to the methodology adopted for collecting the VCR dataset. Zellers et al. [103] instructed workers to provide questions, answers, and rationales by using natural language and object detections **o** (COCO [56] objects). We found that workers used natural language even if the corresponding object detection was available. Additionally, for some data points, we found objects mentioned in the text without a valid object detection in **o**. This may be because the detector used by Zellers et al. [103] is trained on COCO [56], which has only 80 classes.

**Error modes.** We also qualitatively study TAB-VCR's shortcomings by analyzing error modes, as illustrated in Fig. 5. The correct answer is marked with a tick while our prediction is outlined in red. Examples include options with overlapping meaning (Fig. 5(a)). Both the third and the fourth answers have similar meaning which could be accounted for the fact that Zellers et al. [103] automatically curated competing incorrect responses via adversarial matching. Our method misses the 'correct' answer. Another error mode (Fig. 5(b)) is due to objects which aren't present in the image, like the "gloves in a show of flirtatious intent." This could be accounted to the fact that crowd workers were shown context from the video in addition to the image (video caption), which isn't available in the dataset. Also, as highlighted in Fig. 5(c), scenes often offer an ambiguous future, and our model gets some of these cases incorrect.

**Error and grounding.** In Tab. 4, we provide the average number of tags in the query+response for both subtasks. We state this value for the following subsets: (a) all datapoints, (b) datapoints where TAB-VCR was correct, and (c) datapoints where TAB-VCR made errors. Based on this, we infer that our model performs better on datapoints with more tags, *i.e.*, richer association of image and text.

**Error and question types.** In Tab. 6 we show the accuracy of the TAB-VCR model based on question type defined by the corresponding matching patterns. Our model is more error-prone on *why* and *how* questions on the $Q{\rightarrow}A$ subtask, which usually require more complex reasoning.

## 5 Conclusion

We develop a simple yet effective baseline for visual commonsense reasoning. The proposed approach leverages additional object detections to better ground noun-phrases and assigns attributes to current and newly found object groundings. Without an intricate and meticulously designed attention model, we show that the proposed approach outperforms state-of-the-art, despite significantly fewer trainable parameters. We think this simple yet effective baseline and the new noun-phrase grounding can provide the basis for further development of visual commonsense models.

## Acknowledgements

This work is supported in part by NSF under Grant No. 1718221 and MRI #1725729, UIUC, Samsung, 3M, Cisco Systems Inc. (Gift Award CG 1377144) and Adobe. We thank NVIDIA for providing GPUs used for this work and Cisco for access to the Arcetri cluster. The authors thank Prof. Svetlana Lazebnik for insightful discussions and Rowan Zellers for releasing and helping us navigate the VCR dataset & evaluation.

## Footnotes

[1]The dataset also includes information about segmentation masks, which are neither used here nor by previous methods. Data available at: `visualcommonsense.com`

[2]`visualcommonsense.com/leaderboard`

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
