[Supplementary Material · VCR__Dean_Unnat_NeurIPS_19_sup_0109.pdf]

# 6 Supplementary Material for TAB-VCR: Tags and Attributes based Visual Commonsense Reasoning Baselines

We structure the supplementary into two subsections.

1. Details about implementation and training routine, including hyperparamters and design choices.
2. Additional qualitative results including error modes

## 6.1 Implementation and training details

Figure 6: **Accuracy on validation set.** Performance for $Q{\rightarrow}A$ (left) and $QA{\rightarrow}R$ (right) tasks.

As explained in Sec. 3.1, our approach is composed of three components. Here, we provide implementation details for each: (1) BERT: Operates over query and response under consideration. The features of the penultimate layer are extracted for each word. Zellers et al. [103] release these embeddings with the VCR dataset and we use them as is. (2) Joint encoder: As detailed in Sec. 4.3, we assess different variants over the baseline model using two CNN models. The output dimension of each is 2048. The downsample net is a single fully connected layer with input dimension of 2048 (from the image CNN) and an output dimension of 512. We use a bidirectional LSTM with a hidden state dimension of $2 \cdot 256 = 512$. The outputs of which are average pooled. (3) MLP: Our MLP is much slimmer than the one from the R2C model. The pooled query and response representations are concatenated to give a $512 + 512 = 1024$ dimensional input. The MLP has a $512$ dimensional hidden layer and a final output (score) of dimension 1. The threshold for Wu Palmer similarity $k$ is set to $0.95$.

We used the cross-entropy loss function for end-to-end training, Adam optimizer with learning rate $2\mathrm{e}{-4}$, and LR scheduler that reduce the learning rate by half after two consecutive epochs without improvement. We train our model for 30 epochs. We also employ early stopping, *i.e.*, we stop training after 4 consecutive epochs without validation set improvement. Fig. 6 shows validation accuracy for both the subtasks of VCR over the training epochs. We observe the proposed approach to very quickly exceed the results reported by previous state-of-the-art (marked via a solid horizontal black line).

## 6.2 Additional qualitative results

Examples of TAB-VCR performance on the VCR dataset are included in Fig. 7. They supplement the qualitative evaluation in the main paper (Sec. 4.4 & Fig. 3). Our model correctly predicts for each of these examples. Note how our model can ground important words. These are highlighted in **bold**. For instance, for Fig. 7(a), the correct rationale prediction is based on the expression of the **lamp**, which we ground. The lamp wasn't grounded in the original VCR dataset. Similarly grounding the **tag**, and **face** helps answer and reason for the image in Fig. 7(b) and Fig. 7(c). As illustrated via the **couch** in Fig. 7(d), it is interesting that the same noun is present in detections yet not grounded to words in the VCR dataset. This could be accounted to the data collection methodology, as explained in Sec. 4.4 ('explanation of missed tags') of the main paper.

In Fig. 8(a), we provide additional examples to supplement the discussion of error modes in the main paper (Sec. 4.4 & Fig. 5). TAB-VCR gets the question answering subtask (left) incorrect, which we detail next. Once the model knows the correct answer it can correctly reason about it, as evidenced by being correct on the answer justification subtask (right). In Fig. 8(a) both the responses 'Yes, she does like [1]' and 'Yes, she likes him a lot' are very similar, and our model misses the 'correct' response. Since the VCR dataset is composed by an automated adversarial matching, these options could end up being very overlapping and cause these errors. In Fig. 8(b) it is difficult to infer that the the audience are watching a live band play. This could be due to the missing context as video captions aren't available to our models, but were available to workers during dataset collection. In Fig. 8(c) multiple stories could follow the current observation, and TAB-VCR makes errors in examples with ambiguity regarding the future.

(a)

| (Q) Is [1] at summer camp ? | (Q) Is [1] at summer camp ?<br>(A) No , [1] is not at summer camp . |
|---|---|
| ✔ No , [1] is not at summer camp . | [1] is wearing a bikini and there is a pool directly behind her . |
| Yes , [1] is in school . | The formal clothing of [1] and the presence of a wine glass suggest this is not a bathroom or swimming facility , making it unlikely her hair is wet from showering or swimming . |
| No , it ' s the weekend for [0] . | [1] is wearing a bikini and a sash with her hometown written on it . |
| Yes [0] is in italy . | ✔ She is in a bedroom with nice girlish decor and a **lamp** , not a cabin or a tent . |

(b)

| (Q) What are the occupations of [0, 4] ? | (Q) What are the occupations of [0, 4] ?<br>(A) They work at a music store . |
|---|---|
| They are announcers or commentators . | [1, 0] have the stereotypical musician look with long , grungy **hair** and they are in a store that has many guitars on display . |
| ✔ They work at a music store . | [3] is holding a guitar . there is a microphone in between [0, 4] . |
| They are nazi soldiers . | In the old days in small towns , it was common for musicians to set up outside of general stores , which attracted most of the townspeople . |
| [0, 4] are attorneys . | ✔ They are both wearing name **tags**, and there are guitars in the background . |

(c)

| (Q) Is [0] in some sort of danger ? | (Q) Is [0] in some sort of danger ?<br>(A) Yes they seem to be alert and scared . |
|---|---|
| No , [1] is not aware of any danger . | [1] is using an axe as a weapon and [0] is pointing a gun at them to make them stay back . |
| No , [0] is falling all over the place . | [0] is terrified but no one else seems to be in danger . |
| Yes , [1] is in danger . | [1] has a gun up against their head . |
| ✔ Yes they seem to be alert and scared . | ✔ The expression on their **face** is scared or concerned . |

(d)

| (Q) What is [0] doing with [1] ? | (Q) What is [0] doing with [1] ?<br>(A) [0] is carrying [1] to the couch . |
|---|---|
| ✔ [0] is carrying [1] to the **couch** . | [0] looks to be trying to get [1] to stay but he is moving fast to gather his possessions to move out . |
| They are helping [1] get off of a bus . | He is lifting him and carrying him to the exit . |
| [1, 0] decided to dance . | [0] has his arm around [1] ' s shoulder . [1, 0] both look awkward . |
| [0] is letting [1] into the office . | ✔ He is holding her , and he is moving in that direction . |

Figure 7: **Qualitative results.** More examples of the proposed **TAB-VCR** model, which incorporates attributes and augments image-text grounding. The image on the left shows the object detections provided by VCR. The image in the middle shows the attributes predicted by our model and thereby captured in visual features. The image on the right shows *new tags* detected by our proposed method. Below the images are the question answering and answer justification subtasks. The *new tags* are highlighted in **bold**.

(a) Similar Responses

(Q) Does [0] like [1] ?

Yes , she does like [1] .

No , she doesn ' t like him .

She does not know him at all .

✔ Yes , she likes him a lot .

(Q) Does [0] like [1] ?
(A) Yes , she likes him a lot .

She is wearing just a t - **shirt** and grinning up at [1] .

✔ She is leaning very close to him and her expression is happy .

She seems to be enjoying herself while telling him something about shooting a hoop which he is doing .

She ' s watching him and has a proud look on her face .

(b) Missing Context

(Q) Why are [0, 9, 8, 1] , and [2] clapping ?

✔ [0, 9, 8, 1] , and [2] are watching a live band play .

Because they are deciding which performer is the best .

[0, 9, 8, 1] , and [2] are acknowledging what [6, 3] just did on stage .

[0, 9, 8, 1] , and [2] are happy for the couple that just got married .

(Q) Why are [0, 9, 8, 1] , and [2] clapping ?
(A) [0, 9, 8, 1] , and [2] are watching a live band play .

Live music for an audience is better played on a stage where the acoustics can be planned out .

They are in a bar where a live band is playing .

✔ It is common to see live music in some restaurants . clapping is expected after each song is played .

[0, 9, 8, 1] , and [2] are cheering and yelling with wide smiles .

(c) Future Ambiguity

(Q) Will [1] arrive at their destination soon ?

[1] might write someone a ticket .

No , [5, 4] will not be ridden by [1] .

No , they won ' t .

✔ [1] is arriving there now .

(Q) Why are [0, 9, 8, 1] , and [2] clapping ?
(A) [0, 9, 8, 1] , and [2] are watching a live band play .

[1] is surrounded by people at the station , there is a train in the background and people are moving on and off the train .

[1] is in motion and is moving with a quickened pace .

[0] is boarding [4] which is parked outside of a bus station .

✔ [2] can be seen waiting for the carriage .

Figure 8: **Qualitative analysis of error modes.** Responses with (a) similar meaning, (b) lack of context and (c) ambiguity in future actions. Correct answers are marked with ticks and our models incorrect prediction is outlined in red.