[Reviews · NeurIPS 2019]

Reviewer 1



The authors have good results on the VCR dataset. However, the provided dataset is fairly new and it is hard to judge how meaningful these results are. The overall method is not particularly novel, and we already knew from visual question answering works that a richer image representation, e.g., in terms of attributes should help. E.g., "Solving Visual Madlibs with Multiple Cues". A slightly more novel part seems to be tagging.

Reviewer 2



Post-rebuttal: The authors addressed my concern (minor) and I'd recommend the acceptance of the paper as it would provide a strong baseline to the community. General Comments: (G1) L25-L29 need citations [1, 29, 34] (G2) L38: At this point, the readers are left wondering why the proposed model is “Intricate”. Request the authors to add a clear explanation here. (G3) L126: What do the authors mean by “joint encoder is identical”? Is it the same architecture with different parameters? Typos: L42: “to this end we” -> “to this end, we“

Reviewer 3



Many of the gains come from a more thorough approach to analyzing the language (e.g. synsets etc) and new finer labels. A somewhat unfair characterization of this work might be that its gains come primarily from “cleaning up” the data. I’m surprised that there is no benefit from additional fine-tuning of BERT/ResNet and would appreciate a bit more insight into the design choices that were made regarding the modeling (and/or ablations to this end).

[Author Response · NeurIPS 2019]

| Encoder | $Q{\rightarrow}A$ | $QA{\rightarrow}R$ | $Q{\rightarrow}AR$ | Params |
|---|---|---|---|---|
| Shared | 69.69 | 72.18 | 50.52 | 4.7M |
| Unshared | 69.92 | 72.08 | 50.66 | 7.9M |

Table 1: Effect of shared *vs.* unshared parameters in the joint encoder $f(\,\cdot\,;\theta)$ of the TAB-VCR model.

| VCR subtask | Avg. no. of *tags* in query+response | | |
|---|---|---|---|
| | (a) all | (b) correct | (c) errors |
| $Q{\rightarrow}A$ | 2.677 | 2.730 | 2.556 |
| $QA{\rightarrow}R$ | 4.302 | 4.411 | 4.017 |

Table 2: Error analysis as a function of number of tags. Less image-text grounding increases TAB-VCR errors.

We thank all reviewers for their valuable feedback.

R1: Method not particularly novel; we know that a richer image representation helps VQA: While commonsense
reasoning in VCR is evaluated via question answering, VCR data differs from VQA and visual madlibs, even for
the $Q{\rightarrow}A$ subtask. In contrast to VQA, answers are entire sentences. Also, addressing $Q{\rightarrow}A$ and $QA{\rightarrow}R$ (answer
justification) requires "background knowledge about how the world works" [5]. Further, VQA depends on recognition,
which has been largely abstracted away from VCR by providing *tags* (and our *new-tags*).
These differences in data & task necessitate research for novel models and study of their trends. This is substantiated by
poor performance of state-of-the-art VQA models (after retraining) on VCR. More importantly, the VCR performance
trends across models are different from the VQA results. *E.g.*, MUTAN [2] and BUTD [1] achieve 61.04% and 65.05%
on VQAv2 val set (source: [4]) but yield 14.6% and 10.7% on VCR val set (source: [5]).

R1: Overall a good work, but maybe too specific to the given dataset and therefore perhaps not the best fit for NeurIPS:
We think commonsense reasoning is a new aspect of explainability and interpretability of machine learning models. This
has been widely studied by the NeurIPS community, especially in the past years. During the response period, we studied
the impact of *tags* on a new reasoning dataset – GQA [3]. We found that addition of *tags* to our Base+Resnet101
model improves accuracy from 45.85% to 54.96%, on the val set (4 epochs ∼9 hrs. on 2 V100 GPUs & 24 cores).

R2: Earlier reference to citations [1, 29, 34]: We'll refer to the papers early in our revised version, *i.e.*, in L25-L29.
R2: Request to add clear explanation on *intricacy* of existing R2C model in L38: We'll detail the intricacies of the R2C
model: the R2C model has three modules: *grounding*, *contextualization* and *reasoning*. *Grounding* uses a Bidirectional
LSTM to jointly encode language and visual inputs into an encoded query ($\mathbf{q}$) and response ($\mathbf{r}$). *Contextualization*
uses two bilinear attentions: between $\mathbf{r}$ and $\mathbf{q}$ and between $\mathbf{r}$ and object representations $\mathbf{o}$. *Reasoning* concatenates and
feeds the attended query $\hat{\mathbf{q}}$ (from the first attention), the attended object representation $\hat{\mathbf{o}}$ and the encoded response $\mathbf{r}$
into another bidirectional LSTM. The output of this LSTM is again concatenated with the encoded response $\mathbf{r}$ and the
attended query $\hat{\mathbf{q}}$, max pooled and transformed by a multilayer perceptron to predict.

R2: Clarify 'joint encoder is identical' in L126: The joint encoder along with its parameters is shared for processing the
query and response. To validate this design choice, we empirically study that there isn't a significant improvement
(Tab. 1) when using separate weights, which comes at the cost of 3.2M extra trainable parameters. Note that Zellers et
al. also share the encoder for query and response processing. Our design choice makes the comparison fair.

R3: Clarifying fine-tuning and ablations for design choices: *BERT*: For all our models, consistent with [5], the
referenced BERT model is fine-tuned on the VCR dataset. These embeddings of BERT fine-tuned on VCR were
released by the VCR dataset authors: https://github.com/rowanz/r2c/tree/master/data.
*ResNet101*: In Tab. 3 we study the effect of finetuning the last conv block of ResNet101 and the downsample net.
Zellers et al. use row #1. We assess lower learning rates – 0.5x, 0.25x, and 0.125x (#2 to #4). We chose to freeze the
conv block (#5) to reduce trainable parameters by 15M, with slight improvement in performance. By comparing #5 and
#6, we find the downsample net to reduce model size and improve performance. We believe the downsample net (which
trains from scratch) helps adapt the image features to VCR data, removing the need to finetune the last conv block.

R3: Error analysis: In Tab. 4 we show accuracy of the TAB-VCR model based on question type defined by the
corresponding matching patterns. Our model is more error prone on *why* and *how* questions on the $Q{\rightarrow}A$ subtask,
which usually require more complex reasoning. In Tab. 2, we provide average number of tags in the query+response for
the two subtasks for (a) all datapoints (b) datapoints where TAB-VCR was correct (c) datapoints where TAB-VCR
made errors. Our model performs better on datapoints with more tags, i.e., richer association of image and text.

**References:**
[1] P. Anderson, X. He, C. Buehler, D. Teney, M. Johnson, S. Gould, and L. Zhang. Bottom-up and top-down attention for image captioning and visual question
answering. In *Proc. CVPR*, 2018.
[2] H. Ben-younes, R. Cadene, M. Cord, and N. Thome. Mutan: Multimodal tucker fusion for visual question answering. In *Proc. ICCV*, 2017.
[3] D. A. Hudson and C. D. Manning. Gqa: a new dataset for compositional question answering over real-world images. In *Proc. CVPR*, 2019.
[4] M. Shah, X. Chen, M. Rohrbach, and D. Parikh. Cycle-consistency for robust visual question answering. In *Proc. CVPR*, 2019.
[5] R. Zellers, Y. Bisk, A. Farhadi, and Y. Choi. From recognition to cognition: Visual commonsense reasoning. In *Proc. CVPR*, 2019.

| # | Fourth conv block | Downsample net | $Q{\rightarrow}A$ | $QA{\rightarrow}R$ | $Q{\rightarrow}AR$ | Trainable params (mn) |
|---|---|---|---|---|---|---|
| 1 | ■ | ■ | 64.57 | 68.86 | 44.60 | 19.9 |
| 2 | ■ (1/2) | ■ | 64.26 | 68.14 | 44.08 | 19.9 |
| 3 | ■ (1/4) | ■ | 63.11 | 67.73 | 42.87 | 19.9 |
| 4 | ■ (1/8) | ■ | 63.51 | 67.49 | 43.21 | 19.9 |
| 5 | ■ | ■ | **66.47** | **69.22** | **46.45** | **4.9** |
| 6 | ■ | ■ | 65.30 | 69.09 | 45.57 | 7.0 |

Table 3: Ablation for our base model. ■: finetuning and ■: freezing weights of the fourth conv block in ResNet101 image CNN. Presence and absence of downsample net (to project image representation from 2048 to 512) is denoted by ■ and ■.

| Ques. type | Matching patterns | Counts | $Q{\rightarrow}A$ | $QA{\rightarrow}R$ |
|---|---|---|---|---|
| what | what | 10688 | 72.2 | 72.7 |
| why | why | 9395 | 64.7 | 73.2 |
| isn't | is, are, was, were, isn't | 1768 | 75.1 | 66.9 |
| where | where | 1546 | 75.4 | 73.1 |
| how | how | 1350 | 60.4 | 69.6 |
| do | do, did, does | 655 | 71.9 | 68.4 |
| who | who, whom, whose | 556 | 85.1 | 70.1 |
| will | will, would, wouldn't | 307 | 74.3 | 71.0 |

Table 4: Accuracy analysis by question type (with at least 100 counts) of TAB-VCR model. *Why* and *how* questions are most challenging for the $Q{\rightarrow}A$ subtask.

[Meta-Review · NeurIPS 2019]

The paper provides a stronger baseline for the visual commonsense reasoning (VCR) task. All reviewers feel that the work is solid and that providing a strong baseline for the VCR task will help push the field forward.